# A Method for Improving Heat Dissipation and Avoiding Charging Effects for Cavity Silicon-on-Glass Structures

**Junduo Wang** [1,2]**, Yuwei Hu** [1,2]**, Lei Qian** [1,2]**, Yameng Shan** [1,2] **and Wenjiang Shen** [2,]*

[1] School of Nano-Tech and Nano-Bionics, University of Science and Technology of China, Hefei 230026, China; jdwang2017@sinano.ac.cn (J.W.)

[2] Key Laboratory of Nanodevices and Applications, Suzhou Institute of Nano-Tech and Nano-Bionics, Chinese Academy of Sciences, Suzhou 215123, China

* Correspondence: wjshen2011@sinano.ac.cn; Tel.: +86-18550101691

**Abstract:** Anode bonding is a widely used method for fabricating devices with suspended structures, and this approach is often combined with deep reactive-ion etching (DRIE) for releasing the device; however, the DRIE process with a glass substrate can potentially cause two critical issues: heat accumulation on the suspended surface and charging effects resulting from the reflection of charged particles from the glass substrate. In particular, for torsional bars with narrow widths, the heat accumulated on the suspended surface may not dissipate efficiently, leading to photoresist burning and, subsequently, resulting in the fracture of the torsional bars; moreover, once etching is finished through the silicon diaphragm, the glass surface becomes charged, and incoming ions are reflected towards the back of the silicon, resulting in the etching of the back surface. To address these issues, we proposed a method of growing silicon oxide on the back of the device layer. By designing, simulating, and fabricating electrostatic torsional micromirrors with common cavity silicon-on-glass (SOG) structures, we successfully validated the feasibility of this approach. This approach ensures effective heat dissipation on the suspended surface, even when the structure is over-etched for an extended period, and enables the complete etching of torsional bars without adverse effects due to the overheating problem; additionally, the oxide layer can block ions from reaching the glass surface, thus avoiding the charging effect commonly observed in SOG structures during DRIE.

**Keywords:** SOG; DRIE; heat dissipation; charging effect

## 1. Introduction

Micro-electro-mechanical systems (MEMSs) exhibit excellent feasibility in miniaturization sensors due to their small dimension, low power consumption, superior performance, and batch fabrication [1]. MEMS devices have become increasingly important in various fields, such as automation, aeronautics, consumer electronics, defense, industrial manufacturing, medical equipment, life sciences, and telecommunications [2–5]. One popular technique in the manufacture of MEMS devices is anodic bonding, which involves the electrostatic bonding between silicon and glass at low temperatures, offering high bonding strength [6]. Through anodic bonding, predefined cavity silicon-on-glass (SOG) wafers can be created by bonding silicon wafers with specific structures to glass substrates. In MEMS fabrication on SOG wafers, the deep reactive-ion etching (DRIE) technique is widely used due to its effectiveness; however, two significant issues arise during through-silicon etching when using the DRIE process in many predefined cavity-bonded wafer schemes, where heat dissipation and silicon surface damage become critical defects [7,8]. One specific example is the electrostatic torsional micromirror, a widely used micro-optical device in MEMS technology, known for its applications in optical communication, Lidar, laser beam scanning projection, and other related fields [9].

In the pursuit of advanced MEMS devices on cavity SOG wafers, challenges related to heat dissipation during the DRIE process as well as silicon surface damage have emerged

as focal points. This is particularly critical for slender and elongated supporting silicon structures, such as serpent-shaped bars, prevalent in micromirror devices to achieve substantial torsional angles. The continuous heat generated during DRIE accumulates on the silicon membrane surface, predominantly dissipated via narrow torsional bars; however, if the heat accumulation rate exceeds dissipation, detrimental outcomes such as photoresist burning and microstructural damage can occur. Further complicating matters, the charging effect during DRIE—a concern when fabricating devices using cavity SOG—introduces deviations in the trajectory of etched ions, potentially harming the silicon structures [10].

To address these challenges, this study introduces a novel technique for improving the fabrication process of electrostatic micromirrors on cavity SOG wafers. A silicon oxide layer is deposited on the backside of the silicon membrane, within the cavity SOG structure. The oxide layer serves a dual purpose: enhancing heat dissipation during DRIE and acting as an etch-stop layer. The improved heat-dissipation properties of the oxide layer counteract the excessive temperature rise, while its barrier function mitigates the charge effect, safeguarding silicon structures.

The novelty of our approach lies in its departure from conventional methods. Unlike traditional solutions that involve the deposition of a metal layer on the glass surface, electrically connected to the silicon substrate to address etching challenges [11], our method exploits the potential of a silicon oxide layer, optimizing heat dissipation and its etch-stop functions. The distinctiveness of our approach lies in its significant ramifications for micromirror devices—a sector often entangled in the paradoxical struggle between achieving low-voltage operation and accommodating expansive torsional angles. This dilemma typically leads to the abandonment of slender structures for heat dissipation.

In the following sections, we will present the details of the proposed technique and experimental results, highlighting the improvements achieved in the fabrication process of electrostatic micromirrors on cavity SOG wafers. The results demonstrate the effectiveness of this approach in enhancing heat dissipation and preventing silicon surface damage during DRIE, which can have broader implications for other MEMS devices on SOG wafers.

## 2. Micromirror Design

A parallel plate capacitor-based torsional micromirror typically consists of two parts: a torsional mirror substrate and a counter electrode substrate. This is displayed in Figure 1, where g represents the gap between the bottom electrode and the mirror surface, and represents the scanning angle of the micromirror. When a bias voltage is applied to the mirror and counter electrodes, the resulting electrostatic force acts as the driving force to deflect the mirror substrate and change the direction of the light path, thus achieving the modulation of the light.

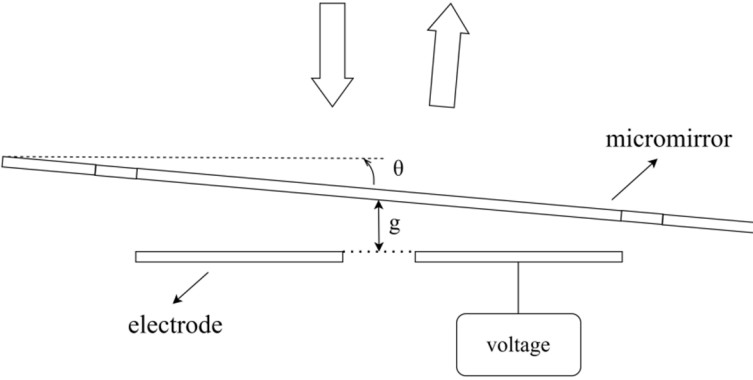

**Figure 1.** The schematic diagram of the micromirror driving principle.

Parallel plate electrostatic micromirrors often require high driving voltages. To minimize the drive voltage, two main approaches are typically considered. One approach involves minimizing the gap between the micromirror and the driving electrode; however,

such a modification would limit the maximum achievable angle of rotation. Another approach is to use compliant torsional bars, which are preferable in reducing the drive voltage. The stiffness of the torsional bar is directly related to the driving voltage. Therefore, optimizing the length and width of the torsional bar can effectively lower the driving voltage. A common method is to utilize the serpentine torsional bars design, which increases the length of the bar and, subsequently, decreases torsional stiffness. For instance, Sadhukhan [12] fabricated a comb-drive electrostatic actuator with a square micromirror measuring 5 mm × 5 mm, while Xiao et al. from Northwestern Polytechnical University produced an array of electrostatic micromirrors [13]. In both cases, they utilized serpentine torsional bars to effectively reduce the driving voltage.

In order to achieve larger scanning angles with lower voltage, the serpentine torsion bar is often made thinner and tightly arranged in parallel, which increases the difficulty of fabrication; however, due to the limitations in the device fabrication process, it is not always feasible to continuously optimize the driving voltage by reducing the width and increasing the length of the torsional bars. As shown in Figure 2, we have designed two micromirrors to compare their mechanical performance and heat-dissipation capabilities with different serpentine torsional bars. Table 1 presents the main design parameters of the two torsional micromirrors. Micromirror No. 1 is characterized by a longer and narrower torsional bar, while micromirror No. 2 maintains the other parameters unchanged. The torsional stiffness of the torsional bar depends on the material properties and dimensions of the torsional bar and can be expressed as [14]:

$$k_\theta = \frac{2Gwt^3}{3L}\left[1 - 0.63\frac{t}{w} + 0.0525\frac{t^5}{w^5}\right] \tag{1}$$

where G is the shear elastic modulus, t is the thickness, w is the width, and L is the length of the torsional bar, respectively.

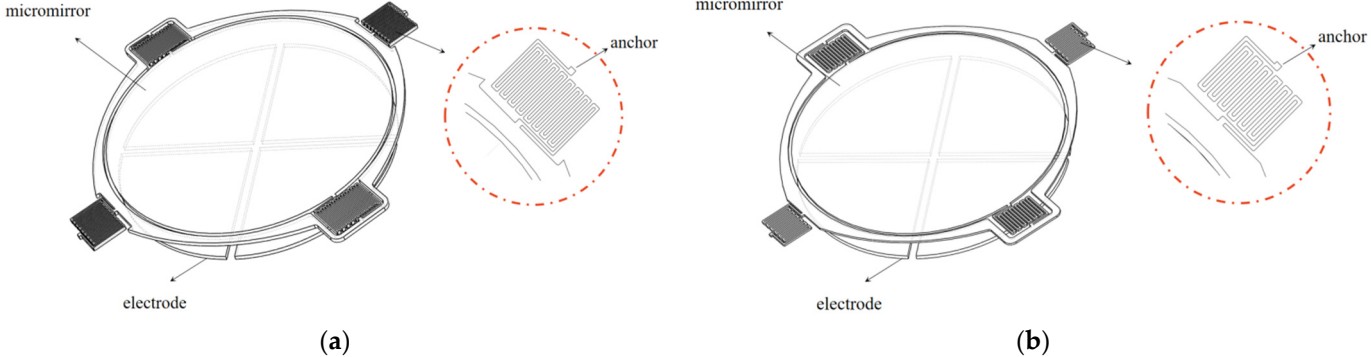

(**a**)            (**b**)

**Figure 2.** Micromirrors with two different torsional bars: (**a**) Micromirror with torsional bars of 3.5 μm width; (**b**) Micromirror with torsional bars of 4 μm width.

**Table 1.** The main design parameters of the two torsional micromirrors.

| Structural Parameters of the Micromirror | No. 1 | No. 2 |
|---|---|---|
| w | 3.5 μm | 4 μm |
| L | 1511 μm | 1170 μm |
| t | 5 μm | 5 μm |
| $k_\theta$ | $5.35 \times 10^{-6}$ N/rad | $6.91 \times 10^{-6}$ N/rad |
| Mirror diameter | 650 μm | 650 μm |
| Mirror thickness | 5 μm | 5 μm |
| Driving electrode thickness | 200 nm | 200 nm |
| g | 35 μm | 35 μm |

The use of a serpentine torsional bar with narrow dimensions during deep silicon etching may result in the burn of the photoresist due to inadequate heat dissipation on the suspended mirror. Upon designing the micromirror and torsional bar dimensions, we employed finite element analysis to predict the heat-dissipation capabilities of the two types of micromirrors, thereby allowing us to refine or adjust the design as necessary.

### 3. Finite Element Analysis Simulation

*3.1. Thermodynamic Simulation*

We conducted thermodynamic simulations on both types of micromirrors using finite element simulation software. In our analysis of heat distribution during the etching process, the selected temperature settings were chosen to closely approximate actual operating conditions. While these temperature values may not precisely replicate the intricate thermal dynamics of the true etching process, our objective is to gauge the overall impact of silicon dioxide on device heat dissipation and etching outcomes. Minor deviations in the chosen temperature settings are unlikely to have a substantial effect on the final results. To elaborate further, the ambient temperature was deliberately set at 343.15 K, emulating the thermal environment within the etching chamber during the DRIE process. Meanwhile, an initial temperature of 303.15 K corresponds to a typical room temperature scenario. Additionally, the glass bottom temperature was consistently maintained at 303.15 K to emulate the connection between the glass substrate and the water cooling system inherent within the etching chamber. Table 2 presents the commonly used etching parameters for deep silicon etching machines, from which the heat flux on the mirror surface during the deep silicon etching process can be estimated to be approximately 2143 W/m$^2$.

**Table 2.** The commonly used etching parameters for deep silicon etching machines.

| $C_4F_8$ | $SF_6$ | $O_2$ | RF Pass Power | ICP Power | Pressure |
|---|---|---|---|---|---|
| 80 sccm | 60 sccm | 5 sccm | 30 W | 600 W | 10 mtorr |

The simulation outcomes, as depicted in Figure 3a,b, reveal that micromirror No. 1 attained a simulated mirror surface temperature of 502.88 K, whereas micromirror No. 2 registered a temperature of 434.63 K. These findings highlight a discernible trend: the heat-dissipation capability of the mirror surface diminishes as the torsional bar of the micromirror assumes longer and narrower dimensions. This observation holds significant implications, as elevated mirror surface temperatures beyond the thermal tolerance of the photoresist can induce photoresist burn and, subsequently, compromise the etching process. In response to this thermal challenge, the strategic integration of an oxide layer into the micromirror's design process emerged as a decisive solution. Positioned at the micromirror's rear, this oxide layer effectively enhances heat transfer during the etching process. Notably, upon reaching the stage of complete penetration through the device layer, the micromirror's structure is etched in its entirety. At this juncture, the oxide layer's presence comes into play, enabling the efficient dissipation of accumulated heat from the mirror's surface. In contrast to relying solely on the narrow connecting bars for heat diffusion, a substantial proportion of the generated heat is effectively channeled through the oxide layer positioned beneath the mirror. This directed heat-flow mechanism ensures the dispersion of heat throughout the micromirror's structure, leading to an enhanced heat-dissipation performance that is particularly significant during the deep reactive-ion etching process. Figure 3c visually demonstrates how the introduction of a silicon dioxide layer as a heat-dissipation medium orchestrates the radial dispersion of heat across the mirror's surface through the silicon dioxide layer. This innovative heat dissipation strategy notably curbs heat accumulation at the torsional bar. When micromirror No. 1 is equipped with a 1000 nm silicon oxide layer as a heat-dissipation layer, the surface temperature is only 306.27 K. Furthermore, employing finite element software, simulations were conducted to explore the influence of oxide layer thickness on heat dissipation. Maintaining a consistent

heat flux, simulations were performed for the same micromirror with different oxide layer thicknesses of 500 nm and 200 nm, culminating in results summarized in Table 3. Clearly, the incorporation of the oxide layer consistently diminishes the mirror temperature under equivalent heat flux conditions, with the oxide layer thickness exhibiting negligible influence on the observed outcomes. This methodical investigation underscores the oxide layer's pivotal role in enhancing heat dissipation, confirming its effectiveness in mitigating thermal concerns within the micromirror's structure.

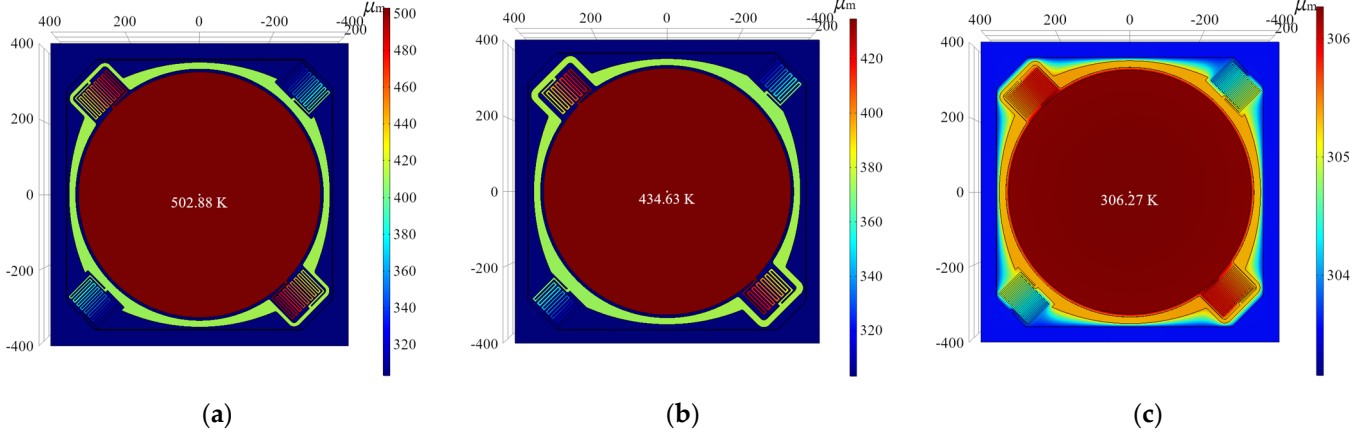

**Figure 3.** Heat distribution on the surface of the mirrors: (**a**) Surface temperature of micromirror No. 1 without oxide layer; (**b**) Surface temperature of micromirror No. 2 without oxide layer; (**c**) Surface temperature of micromirror No. 1 with 1000 nm silicon oxide layer.

**Table 3.** The impact of oxide layer thickness on suspended surface temperature.

| Micromirrors with and without Oxide Layer | Suspended Surface Temperature |
| --- | --- |
| Micromirror No. 2 without oxide layer | 434.63 K |
| Micromirror No. 1 without oxide layer | 502.88 K |
| Micromirror No. 1 with 1000 nm silicon oxide layer | 306.27 K |
| Micromirror No. 1 with 500 nm silicon oxide layer | 308.63 K |
| Micromirror No. 1 with 200 nm silicon oxide layer | 314.75 K |

*3.2. Mechanical Performance Simulation*

Additionally, we conducted a simulation analysis of the mechanical performance of the micromirrors using the solid mechanics module and electrostatic module in the finite element software. When excitation is applied to the bottom electrodes, the mirror surface of the micromirror undergoes motion due to electrostatic attraction. Taking a point at the edge of the micromirror, the mechanical rotation of the micromirror is characterized by its displacement component perpendicular to the bottom electrode. From Figure 4, the observed results reveal that micromirror No. 2 achieves an 8-micrometer displacement with an approximate voltage of 53 V, whereas micromirror No. 1 accomplishes the same displacement with only around 40 V. This demonstrates that increasing the length of the torsional bars and reducing its width can indeed decrease the torsional stiffness coefficient of the torsional bars, thereby reducing the driving voltage required.

By employing finite element analysis software, we conducted comprehensive thermodynamic and mechanical simulations on both micromirror designs. These simulations served as a crucial foundation for our methodology. Drawing insights from these simulations, we made a deliberate choice to proceed with the evaluation of micromirror No. 1. This selection was driven by our intent to rigorously assess whether the presence or absence of silicon dioxide could wield a pivotal influence on DRIE for micromirrors characterized by elongated torsional bars.

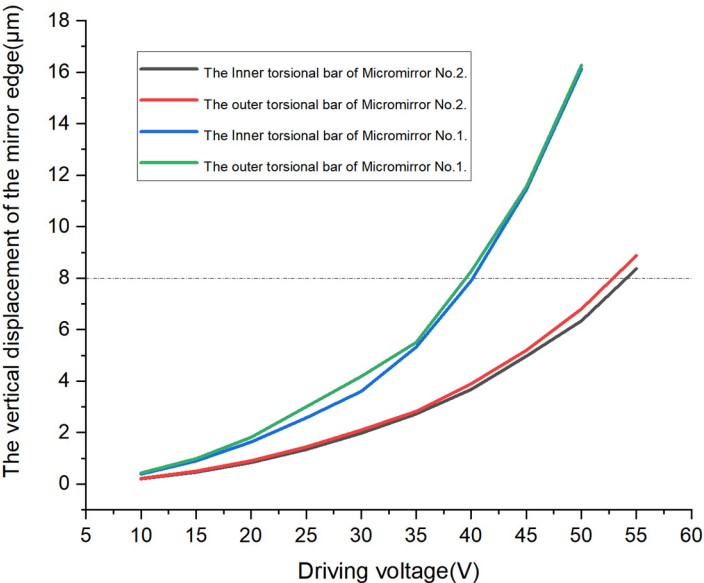

**Figure 4.** Voltage–displacement curves of two types of micromirrors.

## 4. Fabrication

For micromirrors using the parallel plate capacitive driving principle, it is necessary to create a certain gap between the micromirror and the driving electrode to allow for the torsional movement of the mirror. Wafer bonding is a widely used method for achieving the gap. Additionally, compared to silicon-to-silicon direct bonding, silicon-to-glass anodic bonding offers higher bonding strength and lower bonding temperatures, resulting in a more stable bond interface. In this study, the fabrication process adopts silicon-to-glass anodic bonding to achieve the torsional gap in the micromirror. Due to the ease of controlling silicon etching compared to glass, the top silicon layer of a silicon-on-insulator (SOI) substrate is used to create the desired gap and device thickness. This top silicon layer is bonded to a glass substrate with driving electrodes. After removing the bottom silicon layer of the SOI substrate and the buried oxide layer in between, the mirror's reflective coating is applied, followed by etching to release the micromirror structure. This proposed fabrication process is simple, requiring only four lithography steps, eliminating the need for sacrificial and electrical isolation layers. The resulting structure is stable and highly reliable, making it suitable for the production of micromirror arrays and possessing significant practical value.

In this study, we aimed to investigate the effect of a $SiO_2$ heat-dissipation layer on the etching of electrostatic torsional micromirrors. We fabricated two types of micromirrors using anode bonding, one with the heat-dissipation layer and the other without. For the micromirrors with the heat-dissipation layer, the fabrication sequence is shown in Figure 5. The specific micromirror fabrication process is as follows:

(a)　Bottom-Driving-Electrode Fabrication: A 6-inch Borofloat 33 glass wafer, with a thickness of 500 μm, was chosen and meticulously cleaned; subsequently, a Ti (20 nm)/ Au (200 nm) metal layer was deposited on the glass using magnetron sputtering to create the bottom driving electrode;

(b)　SOI Wafer Preparation: A 6-inch SOI wafer with a top silicon layer thickness of 40 μm was chosen and cleaned;

(c)　DRIE for Cavity-Gap Formation: The cavity gap was etched to a depth of 35 μm;

(d)　Heat-Dissipation-Layer Fabrication: A 200 nanometer thick $SiO_2$ layer was grown in the cavity through plasma-enhanced chemical vapor deposition (PECVD) to function as the heat-dissipation layer;

(e)　Anodic Bonding to Form Cavity SOG: Anodic bonding was employed to tightly integrate the processed SOI wafer and the glass wafer under specific conditions;

(f) Removal of SOI's Handle Silicon Layer: The handle silicon layer of the SOI wafer was thinned to around 100 μm and then etched in a 25% Tetramethylammonium hydroxide (TMAH) solution at 85 °C until reaching the oxide layer;

(g) Removal of SOI's BOX Layer: The buried oxide (BOX) layer was removed by immersing it in buffered oxide etch (BOE) solution;

(h) Deposit and Pattern Top Metal Layer: A Ti (20 nm)/Au (200 nm) metal layer was sputtered and then lithography and ion beam etching (IBE) were used to obtain the top electrode and mirror reflective coating;

(i) DRIE for Micromirror Formation: The etching continued until reaching the heat-dissipation layer;

(j) Micromirror Structure Release: The $SiO_2$ heat-dissipation layer was removed using reactive ion etching (RIE), and after laser scribing, the micromirror array bare chips were obtained.

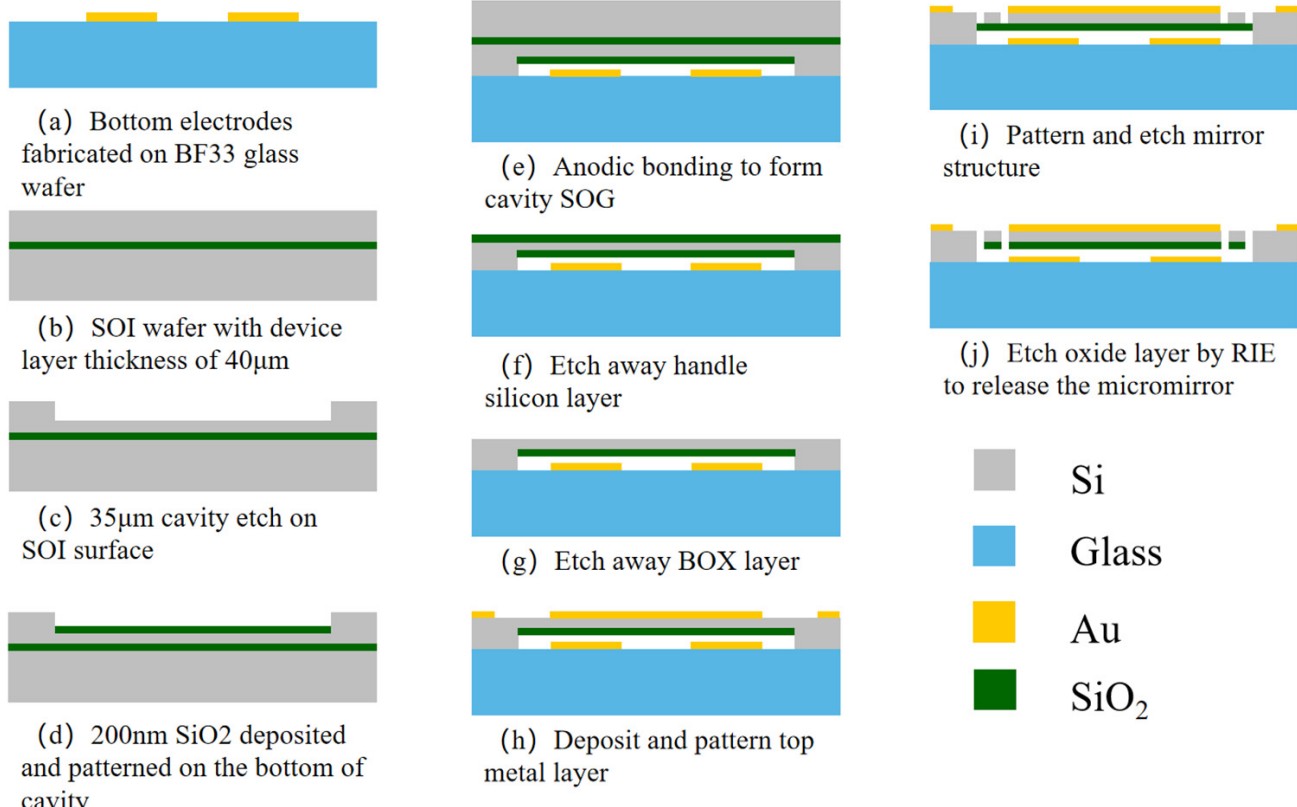

**Figure 5.** Fabrication process of electrostatic torsional micromirrors with heat-dissipation layer.

In contrast, for the micromirrors without the heat-dissipation layer, we omitted the step of growing silicon oxide in the cavity of the SOI device layer. Instead, we directly bonded the SOI with the cavity to the glass with the bottom electrode. These fabrication processes allowed us to compare the effects of the $SiO_2$ heat-dissipation layer on the etching of the micromirrors.

Figure 6a shows the results of forming cavity SOG through anode bonding, with arrows indicating the presence of the oxide layer that is deposited at the bottom of the device layer. Figure 6b shows the mask made using the AZ5214 photoresist prepared for DRIE, as depicted in Figure 5i. The photolithography process was observed to be successful, without any issues arising from poor mask fabrication that could affect the deep silicon etching.

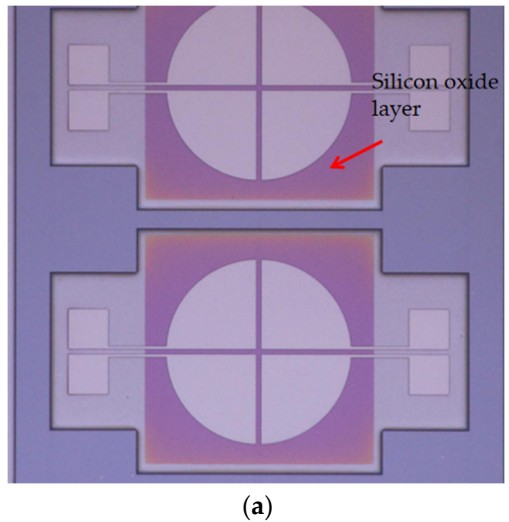
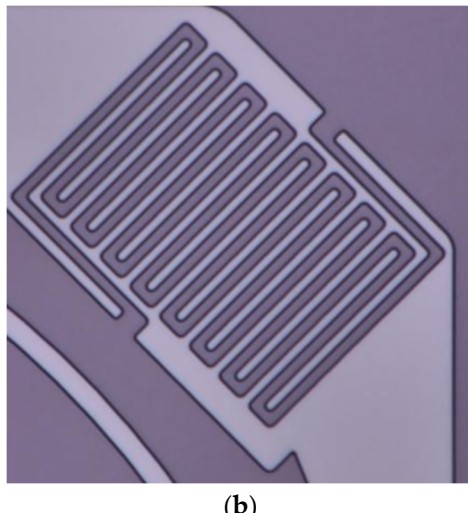

(**a**)                                                                                          (**b**)

**Figure 6.** Photographs of the fabrication process for the electrostatic torsional micromirrors: (**a**) Anodic bonding to form cavity SOG; (**b**) The mask patterned by photoresist (AZ5214).

## 5. Experiments and Discussion

### 5.1. Impact of Silicon Oxide on Etching and Charging Effects

Fabricating electrostatic torsional micromirrors is a complex process that requires overcoming several challenges. One of the challenges is the accumulation of heat on the suspended mirror during continuous etching [15]. This issue can result in photoresist burning and deformation of the polymer in the photoresist, leading to poor etching results and structural damage. As can be seen from Figure 7a, for the micromirrors without the heat-dissipation layer—as in the previous analysis and simulation—a large proportion of the mirror structures crack, and the photoresist on the mirror surfaces is burned. In Figure 7b, it can be observed that the mirror has detached from the outer torsional bar. These observations suggest that the absence of a $SiO_2$ heat-dissipation layer can lead to poor etching results, resulting in photoresist burning and structural damage to the mirrors. This can negatively impact the overall performance of the electrostatic torsional micromirrors and highlights the importance of utilizing a heat-dissipation layer in the fabrication process.

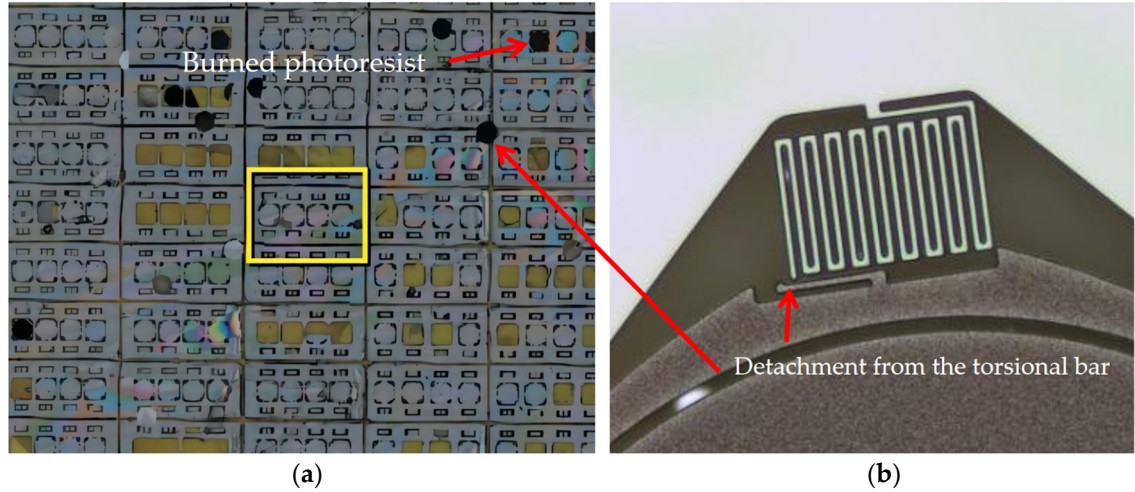

(**a**)                                                                                          (**b**)

**Figure 7.** The electrostatic torsional micromirrors without oxide layer after DRIE: (**a**) The morphology of the micromirrors without oxide layer after DRIE in a wide range; (**b**) Breakage of torsional bars.

In the context of DRIE, it is common for the periphery of the wafer to undergo faster etching compared to the central region; consequently, an extension in etching duration

is often necessary to ensure complete etching of the micromirrors uniformly across the entire wafer. However, when considering micromirrors integrated with a heat-dissipation layer, as depicted in Figure 8a, it is noteworthy that there is an absence of substantial photoresist burning, even when the micromirrors are over-etched; in addition, the enhanced etching outcomes are evident across the entire wafer. Figure 8c provides a comprehensive depiction of the micromirror's post-deep silicon etching state using a profilometer for contour measurement. This analysis underscores the stability and structural integrity of the silicon dioxide layer, as evidenced by the absence of cracking or collapse, even after DRIE. Importantly, Figure 8b illustrates that the torsional bars remain unscathed, signifying that the addition of silicon oxide has proven effective in safeguarding the structural components. A subsequent insight from Figure 8d reveals the enduring integrity of the torsional bars and the successful release of the micromirrors, even after the removal of the silicon oxide layer through reactive ion etching (RIE). This outcome underscores the robustness of the integrated approach.

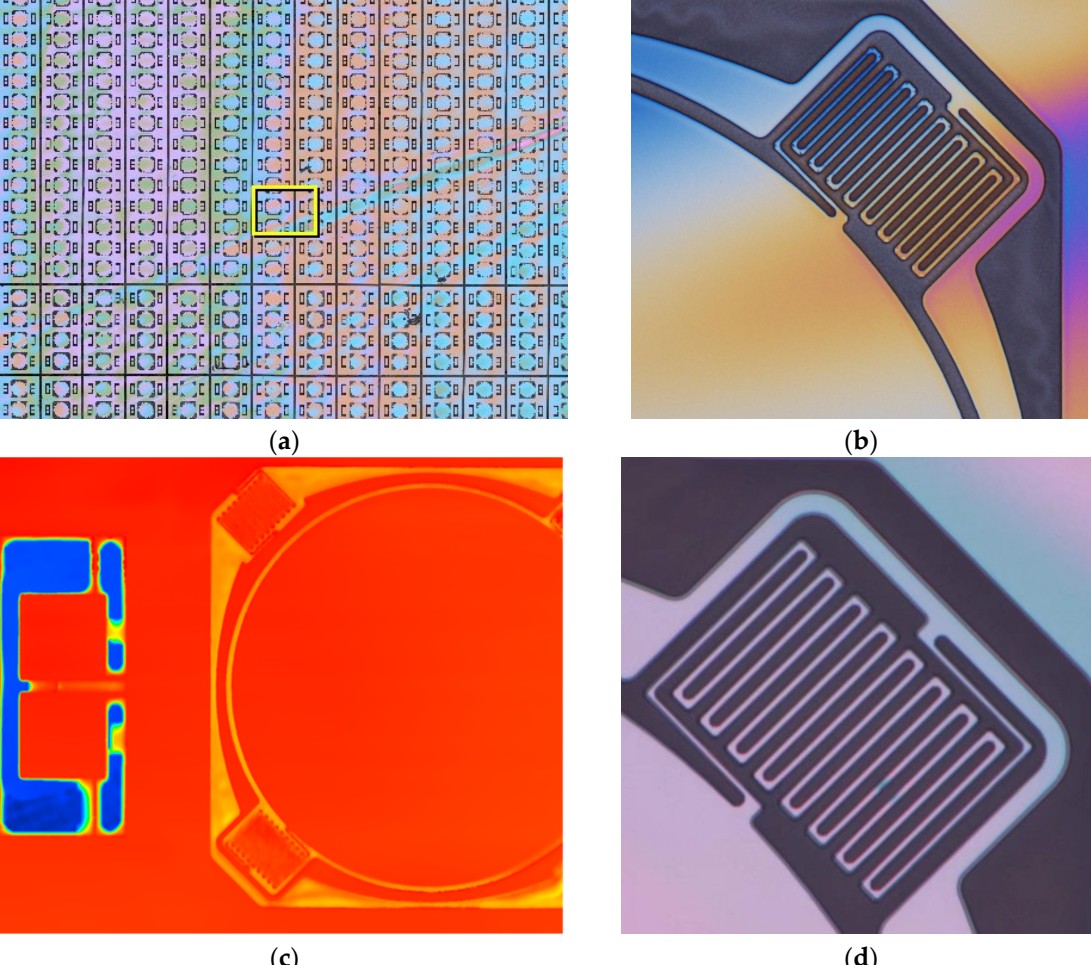

(a)  (b)  (c)  (d)

**Figure 8.** The electrostatic torsional micromirrors with oxide layer after DRIE: (**a**) The morphology of the micromirrors with oxide layer after DRIE in a wide range; (**b**) Torsional bars before oxide layer removal; (**c**) The oxide layer remaining intact after DRIE; (**d**) Torsional bars after oxide layer removal.

Another challenge is the accumulation of charges on the glass surface due to ion bombardment, as illustrated in Figure 9b. This issue can lead to electrostatic deflection of charged ions, resulting in uneven etching or even fracture of the suspended structures [16]. Traditionally, a solution to the problem of electrostatic deflection has been to lay a metal layer on the glass below the suspended bars [17,18]; however, this method is not always suitable for all anode bonding devices, as the electrical connection between the metal layer

and the silicon substrate is difficult to establish, leading to poor bonding quality—particularly for devices such as micromirror arrays with limited bonding areas. As depicted in Figure 9a, visible etching marks were observed on the silicon backside of the micromirror without the heat-dissipation layer, a common phenomenon caused by the charging effects during the etching process; in contrast, as illustrated in Figure 9c, the silicon backside in the micromirror with the heat-dissipation layer was effectively shielded by the oxide layer and, therefore, remained unaffected by the etching ions reflected from the glass substrate. The observed unevenness in the mirror backside was a direct result of the utilization of DRIE to etch the cavity to achieve the appropriate thickness during the fabrication process, as illustrated in Figure 5c.

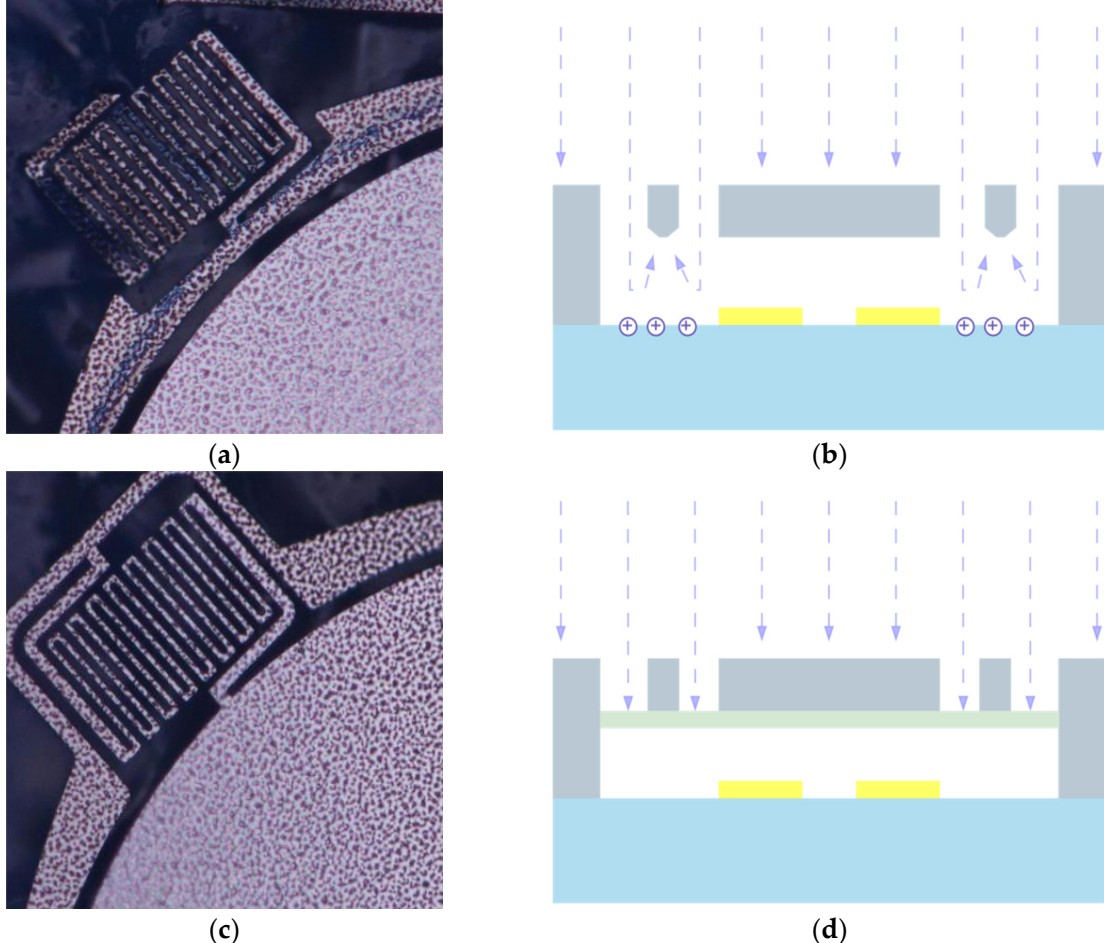

(a)    (b)    (c)    (d)

**Figure 9.** (**a**) The morphology of the backside of the electrostatic torsional micromirror without oxide layer after DRIE; (**b**) Illustration of charging effect; (**c**) The morphology of the backside of the electrostatic torsional micromirror with oxide layer after DRIE and (**d**) Illustration of preventing charging effect.

Figure 10c depicts the successful removal of the oxide layer on the backside of the experimental group device through RIE. Despite undergoing both over-etching in deep silicon etching and RIE etching, the device remained structurally sound, with the torsional bars, made of silicon, remaining undamaged. This result demonstrates the efficacy of the RIE method for the release of micromirrors, which is critical for the successful fabrication of micromirror arrays; furthermore, as shown in Figure 10b, the yield of the devices on the entire wafer was significantly improved and the fabrication of the micromirror array was achieved.

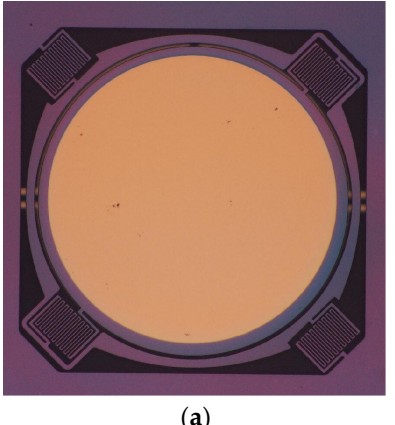
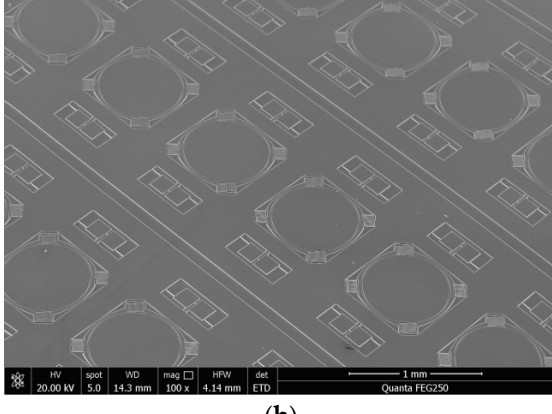

(**a**)  (**b**)

**Figure 10.** Successfully fabricated micromirrors and micromirror arrays: (**a**) Micromirrors with elongated torsional bars; (**b**) SEM image of micromirror arrays.

## 5.2. Mechanical Performance Testing

In this section, we focus on the assessment of the micromirror's mechanical performance, specifically its angular displacement response to applied driving voltages. The objective is to understand the dynamic behavior of the micromirror in response to external stimuli and to validate the consistency between experimental observations and simulation predictions.

The experimental procedure involved directing a laser beam emitted from a laser source onto the center of the mirror's surface. The initial position of the reflected laser beam on the receiving screen was recorded. By subsequently applying a predetermined driving voltage to the micromirror, a controlled angular deflection was induced. This angular displacement prompted a measurable shift in the point of laser beam reflection on the screen. This displacement corresponds to the vertical component of the mirror's edge displacement caused by the mirror's rotation. It allowed us to calculate the corresponding angular deflection. Figure 11 illustrates the results of the mechanical performance testing. Notably, the experimental results exhibit a strong alignment with the outcomes derived from simulations using finite element software, reinforcing the robustness and precision of our analysis.

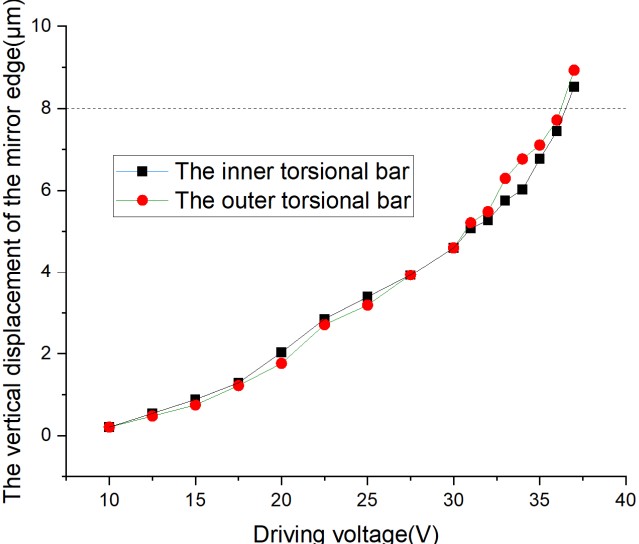

**Figure 11.** Voltage–displacement curves of micromirror.

## 6. Conclusions

The current study aims to tackle the persistent issue of inadequate heat dissipation and charging effects that frequently afflict anode bonding devices during the DIRE process. To overcome these issues, our study involves theoretical and simulation analyses of commonly used electrostatic torsional micromirrors with SOG structures. We propose a novel method of growing silicon oxide on the backside of the device layer and successfully demonstrate the fabrication of micromirror arrays with elongated torsional bars using this approach. Our research has demonstrated that the integration of a silicon oxide heat-dissipation layer has a significant mitigating effect on the occurrence of photoresist burning and mirror fracture during the DIRE process; furthermore, this innovative method holds great potential for enhancing heat dissipation in anode bonding devices, while also effectively preventing etching ions from entering the cavity, thus reducing the risk of device damage. The implications of this method extend beyond the realm of electrostatic torsional mirrors. It offers a promising solution to address heat-dissipation issues for other devices with suspended structures that require DIRE; moreover, it holds great potential in addressing the charging effects commonly seen in the DIRE process.

**Author Contributions:** Conceptualization, J.W.; Methodology, J.W.; Software, J.W.; Validation, J.W.; Formal analysis, J.W.; Investigation, J.W.; Resources, J.W.; Data curation, J.W.; Writing—original draft, J.W.; Writing—review & editing, J.W., Y.H., L.Q. and Y.S.; Visualization, W.S.; Supervision, W.S.; Project administration, W.S.; Funding acquisition, W.S. All authors have read and agreed to the published version of the manuscript.

**Funding:** This research was funded by Research on Shortwave Infrared Chip-based Spectral Detection Technology, grant number 62220106001.

**Institutional Review Board Statement:** Not applicable.

**Informed Consent Statement:** Not applicable.

**Conflicts of Interest:** The authors declare no conflict of interest.

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
