# Peer review of "A Method for Improving Heat Dissipation and Avoiding Charging Effects for Cavity Silicon-on-Glass Structures"

_actuators, doi:10.3390/act12080337_

Round 1

Reviewer 1 Report

The manuscript describes a modified fabrication procedure for silicon structures suspended over the glass substrate. It includes formation of an oxide layer that dissipates heat and protects the structures from charging effect during plasma etching. The proposed technique is demonstrated on a micromirror. Authors provide finite element simulation and experimental results. The paper is interesting for MEMS society. However, it can be published in Actuators after some improvement.

1. Heating of suspended structures and etching of the back side are well known problems in micromachining. The introduction does not provide information on methods for overcoming them. The overview of the current state of the topic is rather short. Thus, it is difficult to understand the scientific novelty. I recommend expanding the introduction and stressing the novelty.

2. In section 3.1, the authors state that silicon oxide layer decreases the mirror temperature, and this effect has poor dependence on the oxide layer thickness. What is the physical reason for heat dissipation, and what dependence on thickness is expected?

3. Section 3.2 is devoted to mechanical performance simulation. It is said in line 159 “These simulation findings are consistent with the theoretical analysis”. However, no theoretical analysis on the mirror deflection and the driving voltage is provided. Furthermore, the simulated deflection is not compared with measurements. I recommend to remove section 3.2, because it gives no useful information.

4. In section 4, line 172 the authors state “Glass, being electrically insulating, allows for natural electrical isolation when fabricating the driving electrodes on it, eliminating the need for additional electrical isolation steps. This simplifies the fabrication process and reduces power consumption during micromirror operation”. Please, explain, how the glass reduces power consumption?

5. In figure 7, it would be useful to mark the cracks, burned photoresist and detachment from the torsion bar.

6. Figure 11a is absent.

Author Response

Dear Reviewer,

Thank you for your valuable feedback and recommendations on our paper. We have carefully reviewed your comments and have made extensive revisions to address each of your points. To facilitate a clear understanding of the changes we've implemented, we have provided a detailed response in the attached document.

Your insights have been instrumental in improving the quality and clarity of our paper, and we are grateful for your thorough review. We kindly request that you review the attached document to see the specific responses we've made to your comments.

Thank you for your time and consideration. We look forward to your continued input.

Best regards,

Junduo Wang

Reviewer 2 Report

J. Wang et al. report their recent study to improve heat dissipation problem and to prevent charging effect for cavity SOG structures. The research is conducted properly. The study and results have practical implication. The article might be considered for publication if the authors can address the following issues:

1.         Can the authors provide the reason to use 343.15K and 303.15K as the initial temperature setting for the simulation? In addition, the authors are suggested to provide the information about the simulation information (e.g. software, model…).

2.         The images in Figure 7(a) and (b) should be enlarged and cropped to show the subject of the interests. (In other words, the devices in the yellow box should be enlarged to become Figure 7(a). The torsion beams in Figure 7(b) should be as large as Figure 7(b).

3.         Figure 8(a) seems not providing enough information. The torsion beams in Figure 8(b)(c)(d) are suggested to be arranged similarly in the pictures so the readers can easily compare the differences.

4.         Figure 9 and Figure 10 can be placed together in one Figure (with 4 sub-figures) so the readers can easily find the difference.

5.         Figure 11(a) is missing.

6.         Can the authors discuss how the results from this study can be applied to other devices with similar fabrication techniques? For instance, the devices with similar torsion bar device at larger scale, or thicker oxide layer…etc.

professional proof-read is suggested.

Author Response

Thank you for your valuable feedback and recommendations on our paper. We have carefully reviewed your comments and have made extensive revisions to address each of your points. To facilitate a clear understanding of the changes we've implemented, we have provided a detailed response in the attached document.

Your insights have been instrumental in improving the quality and clarity of our paper, and we are grateful for your thorough review. We kindly request that you review the attached document to see the specific responses we've made to your comments.

Thank you for your time and consideration. We look forward to your continued input.

Best regards, 

Junduo Wang

Round 2

Reviewer 1 Report

The authors clarified issues mentioned in my report. I recommend acceptance of the paper.